# Genome-wide rare variant score associates with morphological subtypes of autism spectrum disorder

Ada J. S. Chan[1,2], Worrawat Engchuan[1,2], Miriam S. Reuter [1,2,3], Zhuozhi Wang[1,2], Bhooma Thiruvahindrapuram [1,2], Brett Trost [1,2], Thomas Nalpathamkalam[1,2], Carol Negrijn[4], Sylvia Lamoureux[1,2], Giovanna Pellecchia [1,2], Rohan V. Patel [1,2], Wilson W. L. Sung [1,2], Jeffrey R. MacDonald [1,2], Jennifer L. Howe[1,2], Jacob Vorstman [1,5,6], Neal Sondheimer [7,8,9], Nicole Takahashi[10], Judith H. Miles[10], Evdokia Anagnostou[9,11], Kristiina Tammimies [12], Mehdi Zarrei[1,2], Daniele Merico [1,13], Dimitri J. Stavropoulos[14,15], Ryan K. C. Yuen [1,2,7], Bridget A. Fernandez [4,16,17] ✉ & Stephen W. Scherer [1,2,7,18] ✉

Defining different genetic subtypes of autism spectrum disorder (ASD) can enable the prediction of developmental outcomes. Based on minor physical and major congenital anomalies, we categorize 325 Canadian children with ASD into dysmorphic and nondysmorphic subgroups. We develop a method for calculating a patient-level, genome-wide rare variant score (GRVS) from whole-genome sequencing (WGS) data. GRVS is a sum of the number of variants in morphology-associated coding and non-coding regions, weighted by their effect sizes. Probands with dysmorphic ASD have a significantly higher GRVS compared to those with nondysmorphic ASD ($P = 0.03$). Using the polygenic transmission disequilibrium test, we observe an over-transmission of ASD-associated common variants in nondysmorphic ASD probands ($P = 2.9 \times 10^{-3}$). These findings replicate using WGS data from 442 ASD probands with accompanying morphology data from the Simons Simplex Collection. Our results provide support for an alternative genomic classification of ASD subgroups using morphology data, which may inform intervention protocols.

Autism spectrum disorder (ASD), which is diagnosed on the basis of behavioral assessments that reveal social communication deficits and repetitive behaviors, is often associated with traits, including major congenital anomalies (MCAs), minor physical anomalies (MPAs)[1,2], and intellectual disability[3–5]. Increasingly, penetrant variants of diagnostic value[6,7] and lesser impact common variants are being implicated in the etiology of ASD[4,8].

Autistic individuals who are more dysmorphic (complex ASD) tend to have lower intelligence quotients (IQ) and more brain and other major congenital anomalies[9,10] compared with those who are less dysmorphic (essential ASD), leading to poorer developmental outcomes. Individuals with complex ASD are also less likely to have a family history of ASD, suggesting that morphological subtypes can reveal informative genetic differences among ASD subgroups[9].

Genetic liability to ASD can be quantified using a polygenic risk score (PRS), which is a weighted sum of ASD-associated common variants, using effect sizes drawn from genome-wide association studies[11]. A similar score for rare variants remains to be established.

**Fig. 1 | Project workflow.** Summary of phenotype stratification, whole-genome sequencing workflow, and genomic analyses performed in this study. ASD autism spectrum disorder, ADM autism dysmorphology measure, CNVs copy-number variants, SNVs single-nucleotide variants, indels insertions and deletions, ERDS estimation by read depth with single-nucleotide variants, GATK-HC Genome Analysis Toolkit-Haplotype Caller, SNPs single-nucleotide polymorphisms, ACMG American College of Medical Genetics and Genomics, IQ intelligence quotient. *Unaffected siblings were used for GRVS and PRS analyses. **Excluding samples with false negative ADM-defined nondysmorphic ASD. We also included only samples sequenced on Illumina platforms to be consistent with the replication

cohort. For variant calling, on average per sample, we detected ~3.7 million SNPs, 36,514 rare single-nucleotide variants (SNVs), 4113 small insertions and deletions (indels), 13 rare copy-number variants (CNVs), 390 rare SVs, 73.4 de novo SNVs, 7.3 de novo indels, and 0.1 de novo CNVs (Supplementary Data 2). Experimental validation rates were 94.8%, 85.7%, and 87.5%, respectively, for de novo SNVs, indels, and CNVs (Supplementary Data 3 and 4). Using GRVS, we were able to quantify and validate the contribution of morphology-associated, rare sequence-level and copy-number variants to morphological ASD subtypes. While we can call other SVs from the WGS, there needs to be higher-quality data before these can be effectively incorporated into GRVS.

Rare variant studies use burden analyses to compare the frequency of rare variants, equally weighted, between cases and controls or among ASD subtypes[3,12,13]. Quadratic tests have also been used in rare variant association tests and typically weigh variants by minor allele frequency[14,15]. Two recent studies have calculated a rare variant risk score based on the number of variants overlapping specific genes[16] or a score that weighs sequence variants by variant type (i.e., loss-of-function and missense variants) and inheritance[17]. However, additional variant types, such as copy-number variants, remain to be included in weighted scores. Moreover, the effect size of a variant not only depends on the variant type, but also on the function, expression, or disease association of the gene and should also be considered in the rare variant score.

Here, from two independent cohorts, we use whole-genome sequences (WGS) and detailed clinical morphology data to: (1)

develop a genome-wide rare variant score (GRVS) to measure the relationship between rare variants and morphology, and (2) examine the contribution of rare and common variants in morphological ASD subtypes (Fig. 1 and Supplementary Fig. 1). We show that probands with dysmorphic ASD have a significantly higher GRVS compared to those with nondysmorphic ASD, and that there is an over-transmission of ASD-associated common variants in nondysmorphic ASD probands.

## Results and discussion
### Stratification of discovery cohort by morphological anomalies
For our discovery cohort, we used a population-based sample of 325 unrelated children with Autism Diagnostic Observation Scale (ADOS)-confirmed ASD. Following clinical examination, a total morphology score was assigned to each case based on the number of MPAs and

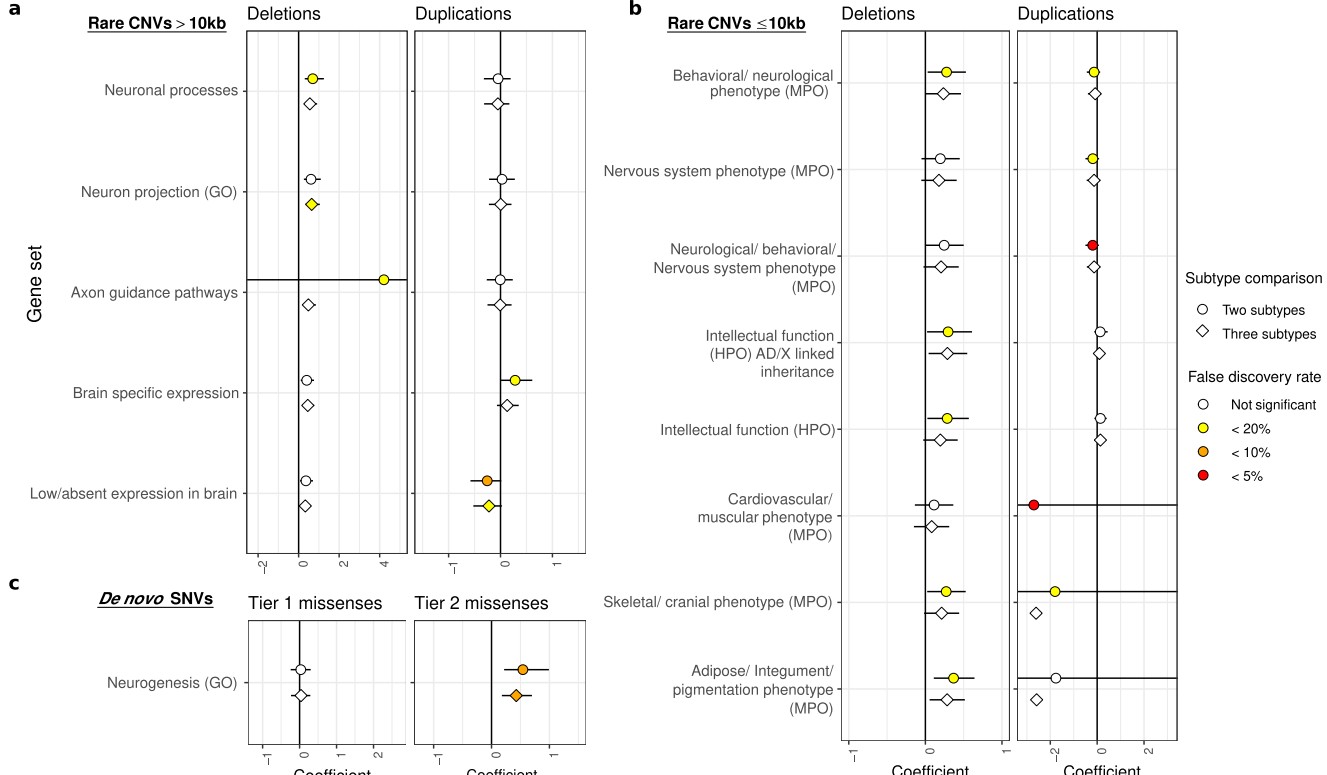

**Fig. 2 | Gene sets for which de novo and rare coding variants are significantly more prevalent in some subtypes of ASD.** We define events as (**a**, **b**) genes impacted by CNVs or (**c**) as variants for SNVs and indels. The coefficient is the relationship between the number of events in each gene set and the ASD subtypes; it reflects the effect size of a variant type and gene set among different ASD subtypes. Positive coefficients indicate more events in individuals with ASD and more dysmorphic features; negative coefficients indicate more events in individuals with ASD and fewer dysmorphic features. We show only gene sets for which (**a**, **b**) rare CNVs (n = 325 samples), or (**c**) de novo missense variants (n = 235 samples) are significantly more prevalent in different subtypes of ASD. Tier 1 and 2 missense variants consist of all or only predicted damaging missense variants, respectively, as defined in ref. 25. Symbol shapes indicate the subtype comparisons that were conducted for each combination of the gene set and variant type. Two-sided likelihood ratio tests were performed with permutation-based FDR for the multiple-testing correction. Two subtype comparison = nondysmorphic vs. dysmorphic ASD. Three subtype comparison = essential vs. equivocal vs. complex ASD. Colored shapes indicate significant signals after multiple test correction by permutation-based FDR, where yellow, orange, and red, indicate permutation-based FDR < 20%, 10% and 5%, respectively. The data points (the center) indicate the estimated coefficient, while error bars indicate 95% confidence intervals of the estimated coefficient. Source data are provided as a Source Data file.

MCAs[9,10]. The cohort was then stratified into three subtypes of increasing morphologic severity: 187 essential ASD (57.5%), 57 equivocal ASD (17.5%), and 81 complex ASD (24.9%) (Supplementary Data 1). We further stratified these samples into two subtypes by combining complex and equivocal ASD into a single dysmorphic ASD grouping to increase power and redefining essential ASD as nondysmorphic ASD.

**Clinically significant rare variant analysis from WGS**
We performed WGS on 795 genomes (325 probands and 470 parents), which are new to this publication, and detected all classes of variation (SNV, indel, CNV and structural variants (SVs) (Fig. 1 and Supplementary Data 2–4). Using the American College of Medical Genetics and Genomics guidelines[18,19], we identified a total of 46 clinically significant variants (CSVs) in 46 of 325 probands (14.1%) (Supplementary Data 5–7). The proportion of dysmorphic ASD cases with a CSV (25.4%; 35/138) was significantly higher than for nondysmorphic ASD (5.9%, 11/187) ($P = 6.6 \times 10^{-7}$, odds ratio = 5.4, 95% confidence intervals = [2.6, 12.3], one-sided Fisher's test), consistent with our previous findings[10]. We also identified 29 variants of uncertain significance (VUS) in 26 probands that were of interest, including tandem repeat expansions in previously reported ASD candidate loci[20]; three probands each had two VUS (Supplementary Data 5–7 and Supplementary Note 1).

**Rare variant burden and enrichment analyses**
To further investigate the contribution of rare variants among morphological ASD subtypes, we first conducted a rare variant burden analysis and multiple test correction using the Benjamini Hochberg approach (BH-FDR) (see "Methods"). We found a significantly higher prevalence of rare coding deletions >10 kb in probands with more dysmorphic features (Deviance statistics (degrees of freedom = 1) = 9.97, $P = 1.59 \times 10^{-3}$, Beta = 0.50, 95% confidence intervals = [0.16, 1.00], and BH-FDR = $3.17 \times 10^{-3}$, two-sided likelihood ratio test, Supplementary Data 8). Rare coding duplications >10 kb and ≤10 kb, genic deletions ≤10 kb, loss-of-function (LoF), and missense variants were not significantly different among subtypes (Supplementary Data 8).

We then performed enrichment and burden analyses to identify gene sets or noncoding regions, respectively, that were differentially affected by rare or de novo variants between the morphological ASD subtypes. The 71 gene sets and noncoding regions studied have been previously associated with ASD or developmental disorders[13,21–27]. After multiple-testing correction (permutation-based false discovery rate (FDR) < 20%), 20 significant gene sets or noncoding regions were identified (Supplementary Data 9 and 10). We observed that probands with dysmorphic features had higher burdens of deletions and missense variants impacting genes responsible for various neuronal functions and duplications >10 kb impacting brain-expressed genes (Fig. 2 and Supplementary Data 9). Dysmorphic probands also had a

significantly higher prevalence of rare deletions ≤10 kb overlapping promoters of long noncoding genes, and duplications (larger and smaller than 10 kb) overlapping active brain enhancers (Supplementary Fig. 3 and Supplementary Data 10).

## Calculation of GRVS for each proband in the discovery cohort

We then tested the collective contribution of rare variants in morphology-associated regions, while considering the effect size of each variant, which varies depending on the variant type and morphology-associated region. We developed a GRVS for each proband, which is a weighted sum of the number of rare variants in morphology-associated regions identified from gene set enrichment and noncoding burden tests (Supplementary Data 11). We weighed the number of rare variants in each morphology-associated region as well as the variant type (i.e., coding or noncoding deletions and duplications >10 kb or ≤10 kb, loss-of-function variants, missense variants, and noncoding SNVs and indels) using the coefficients from logistic regression models.

To calculate GRVSs for each proband in the discovery cohort, we used a tenfold cross-validation strategy to reduce overfitting (Supplementary Fig. 1a). We used Nagelkerke's $R^2$ to determine the optimal $P$ value threshold ($P < 0.1$) to identify morphology-associated regions (Supplementary Fig. 4a and "Methods"). GRVS can be calculated for probands regardless of whether their parents have been sequenced. However, there would be a systematic difference in GRVSs in this cohort if all probands were used because those probands whose parents have been sequenced would include scores from de novo variants, whereas those without sequenced parents would not have de novo variant scores. To avoid this, GRVS was calculated only for probands with two sequenced parents ($n = 235$) (Fig. 3a and Supplementary Data 12).

Probands with dysmorphic ASD had significantly higher average GRVSs than those with nondysmorphic ASD ($P = 0.03$, one-sided Wilcoxon rank-sum test) (Fig. 3b). Most probands (96.6%, 227/235) had more than one variant impacting morphology-associated regions (Supplementary Data 12). Rare coding CNVs had the highest effect size; rare noncoding SNVs and indels had the lowest (Fig. 3c and Supplementary Data 11). When we excluded the probands with high impact variants (i.e., deletions or LoFs impacting LoF intolerant genes or missense variants with the missense badness, PolyPhen-2, and constraint[28] (MPC) score > 2) in 183 ASD genes[26], those with dysmorphic ASD still had higher average GRVSs than those with nondysmorphic ASD with a trend toward significance ($P = 0.07$, one-sided Wilcoxon rank-sum test). These findings suggest that variants in morphology-associated regions that are not known to have high impact also contribute to morphological outcomes in ASD.

## Contribution of clinically significant variants to GRVS

Using the GRVS formula, we calculated a score for CSVs that overlapped an ASD-relevant, morphology-associated region (so that effect size was available for calculation) and that occurred in probands with sequencing data from both parents, of which 17 of the 46 CSVs met these criteria. No score was calculated for the remaining 29 variants because 15 were identified in probands where both parents were not available for sequencing, and 14 variants were not located in or encompassed by one of the 20 morphology-associated regions ("Methods"). In 47% of samples with CSV scores (8/17 probands, Supplementary Data 13), CSVs contributed >50% of the total GRVS. When we excluded the probands with CSVs, those with dysmorphic ASD still had significantly higher average GRVSs than those with nondysmorphic ASD ($P = 0.044$, one-sided Wilcoxon rank-sum test, Fig. 3b). These findings suggest that variants in morphology-associated regions that are not CSVs also significantly contribute to morphological outcomes in ASD.

## Analysis of common variants using polygenic TDT

To explore the contribution of common (minor allele frequency >0.05) ASD-associated variants in ASD subtypes, we calculated polygenic risk scores (PRS) for ASD and body mass index (BMI)[8] ("Methods" and Supplementary Data 12). We then compared these scores across the morphologic groups using the polygenic transmission disequilibrium test (pTDT)[8], which compares the PRS of the proband to the parents' mean PRS. We found a significant overtransmission of common ASD-associated variants in probands with nondysmorphic ASD ($P = 2.9 \times 10^{-3}$, one-sided Welch's $t$-test) and no significant overtransmission in probands with dysmorphic ASD ($P = 0.3$) (Fig. 4). PRS for BMI was selected as a negative control because there is no genetic correlation between BMI and ASD[29], and we did not find overtransmission of PRS for BMI in either subtype (Fig. 4).

## IQ correlations

IQ is often negatively correlated with the burden of rare variants[3,4,13,30,31]. We therefore examined our probands with dysmorphic ASD and determined they had a significantly lower mean IQ compared to nondysmorphic ASD ($P = 0.013$, one-sided Welch's $t$-test, Fig. 5a and Supplementary Data 12). Probands with a CSV had significantly lower IQ compared to probands without a CSV ($P = 2.2 \times 10^{-4}$, one-sided Welch's $t$-test, Fig. 5b). However, IQ was not significantly correlated with GRVS (rho = −0.042, $P = 0.64$, Fig. 5c) or PRS (rho = −0.15, $P = 0.12$, Fig. 5d).

## Clinical reclassification of discovery cohort for comparison to replication cohort

We repeated our analysis on a replication cohort of relevant samples from the Simons Simplex Collection (442 ADOS-confirmed affected probands and 355 unaffected siblings)[32]. The affected probands had been categorized into two morphological subtypes (400 nondysmorphic and 42 dysmorphic cases)[32] using the Autism Dysmorphology Measure (ADM)[33]. In contrast to the discovery cohort, the SSC probands were classified by targeted physical examinations performed by individuals without expert training in dysmorphogy, and the classification did not incorporate the presence or absence of major congenital anomalies. To compare the two cohorts, we reclassified a subset of the original discovery cohort based on minor anomalies alone using the ADM algorithm (203 nondysmorphic and 73 dysmorphic cases, "Methods"). We calculated new GRVSs for the ADM-reclassified discovery cohort using a tenfold cross-validation approach (143 nondysmorphic and 48 dysmorphic cases met criteria for inclusion in this analysis, Supplementary Fig. 1a, Supplementary Data 12 and 14, and "Methods"). We used Nagelkerke's $R^2$ to determine the optimal $P$ value threshold and identified 35 morphology-associated regions, which largely overlapped with our original analysis (Supplementary Fig. 4b). The morphology-associated regions ($P < 0.1$, Supplementary Data 15) identified in the reclassified discovery cohort were used to calculate GRVSs for the replication cohort (Supplementary Fig. 1b, Supplementary Data 16, and "Methods").

## GRVS analyses in discovery and replication cohorts

In both cohorts, probands with ADM-defined dysmorphic ASD had significantly higher GRVSs ($P_{discovery} = 4.7 \times 10^{-7}$ and $P_{replication} = 5.8 \times 10^{-3}$, one-sided Wilcoxon rank-sum test, Fig. 6a) and yield of CSVs ($P_{discovery} = 2.7 \times 10^{-7}$ and $P_{replication} = 2.1 \times 10^{-3}$, one-sided Wilcoxon rank-sum test, Fig. 6b and Supplementary Data 17 and 18) compared to ADM-defined nondysmorphic ASD, consistent with our findings using the gold-standard dysmorphology classification. In the replication cohort, unaffected siblings had a significantly lower GRVS compared to ADM-defined dysmorphic ASD ($P = 6.7 \times 10^{-3}$ one-sided Wilcoxon rank-sum test) but did not have a significantly lower GRVS compared to ADM-defined nondysmorphic ASD (Fig. 6a). Furthermore, unaffected siblings

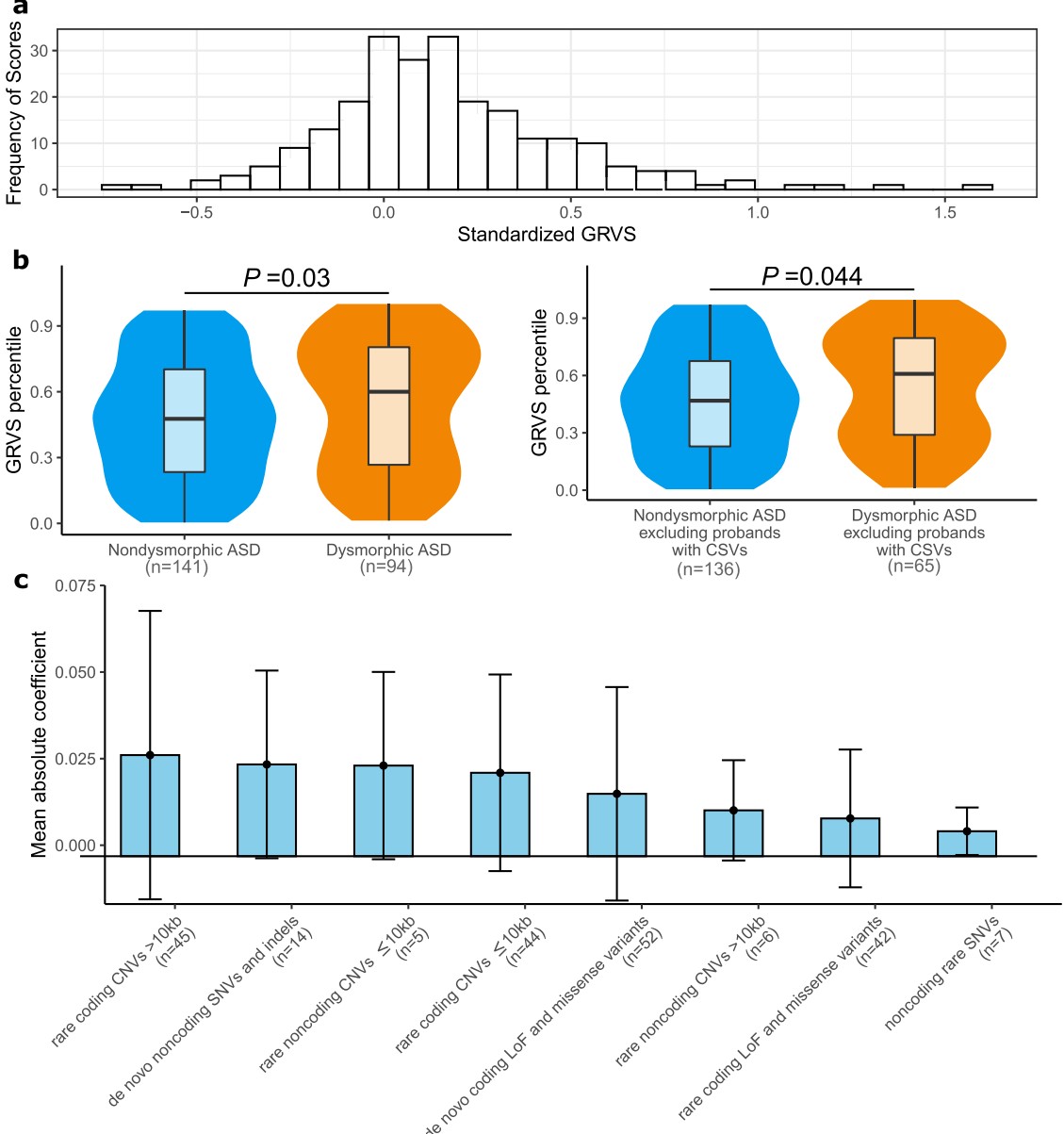

**Fig. 3 | Genome-wide rare variant score in ASD subtypes.** Events are comprised of variants for SNVs and indels or genes impacted by CNVs. For each sample, the GRVS is the sum of rare and de novo events in morphology-associated regions, weighted by effect size (estimated from the coefficients in the regression model). GRVSs were generated 30 times for each sample (see "Methods"), yielding an average score and average number of variants. CSVs, clinically significant variants. **a** Distribution of standardized GRVS for the discovery cohort (*n* = 235). **b** GRVSs for the whole cohort (left plot, *n* = 235) or the whole cohort excluding the 17 probands with clinically significant variants (right plot, *n* = 218), were ordered and ranked by percentile. Note that while 46 probands in the discovery cohort (*n* = 325) had CVSs, only 17 of them had two sequenced parents meeting inclusion criterion for the GRVS group (*n* = 235). The minima and maxima of box plots indicate 3× the interquartile range

deviated scores from the median, and the center indicates the median of the score percentiles. Violin plots show the distributions of the samples' GRVS percentiles; box plots contained within show the median and quartiles of the percentiles for each subtype. *P* values denote the probability that the GRVS in dysmorphic ASD is not greater than nondysmorphic ASD (one-sided, Wilcoxon rank-sum test). **c** Rare variants have different effect sizes. The mean coefficient reflects the effect size of a variant type. Coefficients of deletions and duplications of the same size bin were averaged together. Coefficients of predicted LoF variants, missense variants, and predicted damaging missense variants were averaged together. Error bars indicate mean ± standard deviation. The number of morphology-associated regions for each variant type is indicated the *y* axis with "*n* =". Source Data are provided as a Source Data file.

of nondysmorphic probands did not have a significantly lower GRVS compared unaffected siblings of dysmorphic probands (*P* = 0.19, one-sided Wilcoxon rank-sum test). We repeated the GRVS analyses separately for European and non-European subsets. Our finding stayed the same for both subsets, where probands with ADM-defined dysmorphic ASD had significantly higher GRVSs when compared to ADM-defined nondysmorphic ASD ($P_{discovery\_eur} = 2.3 \times 10^{-5}$ and $P_{replication\_eur} = 1.3 \times 10^{-3}$, $P_{replication\_noneur} = 3.1 \times 10^{-4}$, one-sided Wilcoxon rank-sum test) or unaffected siblings ($P_{replication\_eur} = 8.2 \times 10^{-4}$, $P_{replication\_noneur} =$

$1.5 \times 10^{-4}$) (Supplementary Fig. 7 and Supplementary Note 1). This result suggests that GRVS is quite robust to a population bias and could be applied across different populations.

**Common variant analyses in discovery and replication cohorts**
In both cohorts, we also found a significant overtransmission of common ASD-associated SNPs in ADM-defined nondysmorphic ASD ($P_{discovery} = 6.7 \times 10^{-3}$ and $P_{replication} = 6.3 \times 10^{-3}$, one-sided Wilcoxon rank-sum test, Fig. 6c). In results similar to ref. 8, we did not observe

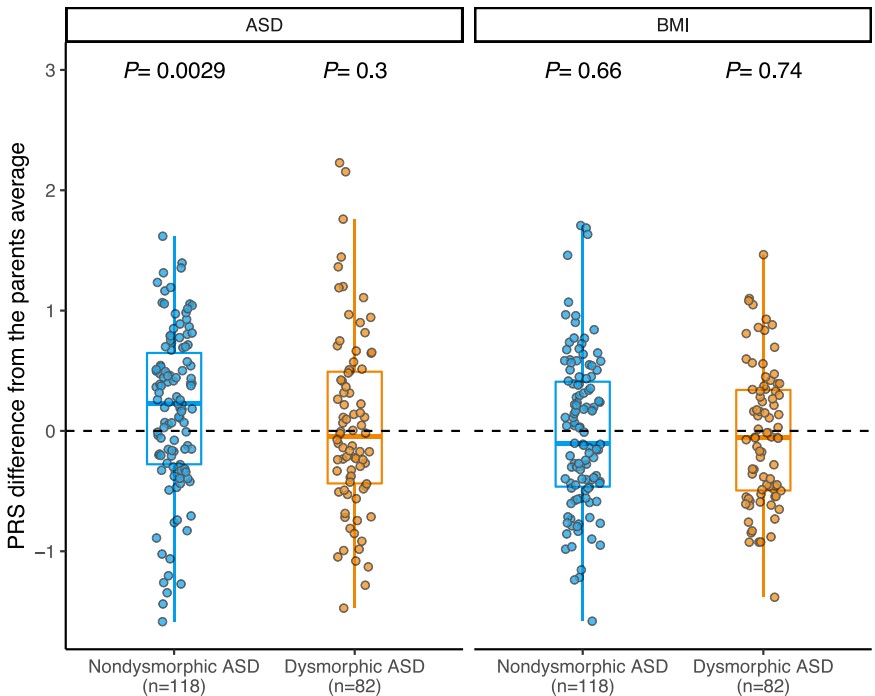

**Fig. 4 | Inheritance of polygenic risk for ASD and BMI in morphologic subtypes.** Differences in polygenic risk score (PRS) for ASD and BMI between participants and their respective mid-parent score. Box plots depict the median and quartiles of polygenic transmission disequilibrium test (pTDT) deviation, and the minima and maxima of box plots indicate 3× the interquartile range-deviated pTDT deviations from the median. Dots represent pTDT deviations of participants. *P* values for each subgroup indicate the probability that the mean of the pTDT deviation distribution is not greater than zero (one-sided Welch's *t*-test was performed assuming pTDT deviation to be greater than zero), as depicted by the dotted line. Source data are provided as a Source Data file.

overtransmission in unaffected siblings in the replication cohort ($P = 0.88$, one-sided Wilcoxon rank-sum test). Although we did not find a correlation between PRS and GRVS in the discovery cohort, a significant positive correlation between PRS and GRVS was observed in the replication cohort (Spearman's rho = 0.11, $P = 0.038$) (Supplementary Fig. 6 and Supplementary Data 12 and 16).

### IQ comparisons

Individuals with ADM-defined dysmorphic ASD or with CSVs had a significantly lower IQ compared to ADM-defined nondysmorphic ASD or those without CSVs, respectively (Supplementary Fig. 5a, b and Supplementary Data 12, 14–16). Although there was no correlation between IQ and GRVS in the discovery cohort when the subtype classification was done by either gold-standard dysmorphology examination (rho = −0.042, $P = 0.64$, Fig. 5c) or using the ADM (rho = 0.081, $P = 0.42$, Supplementary Fig. 5c), a significant negative correlation was found in the replication cohort (rho = −0.12, $P = 0.016$, Supplementary Fig. 5c). We did not find significant correlations between IQ and PRS (Fig. 6d and Supplementary Fig. 5d), in either cohort (Supplementary Fig. 6 and Supplementary Data 12 and 16).

Differences in the correlation between GRVS and IQ between the cohorts might be attributable to differences in ascertainment. The discovery cohort was assembled using a population-based recruitment strategy, and the average IQ of the cohort is 105, similar to the population average of 100. In contrast, individuals with comorbid ID or low IQ are found in SSC[34], consistent with the replication cohort having a significantly lower IQ compared to the discovery cohort (mean IQ$_{discovery}$ = 105 ± 23, mean IQ$_{replication}$ = 82 ± 27, $P = 1.1 × 10^{-21}$, two-sided Welch's *t*-test). Inconsistent findings between ASD cohorts have also been observed when examining sex differences in IQ[35], where findings from cohorts with specific selection criteria (e.g., simplex families) may not be generalizable to the broader ASD population.

### Conclusions

Our data suggest that while both dysmorphic and nondysmorphic ASD demonstrate overtransmission of common ASD-associated variants, there is a significantly higher burden of rare variants in dysmorphic ASD than nondysmorphic ASD. GRVS methods may add further specificity to identifying clinically informative endophenotypes but exquisitely phenotyped cohorts will be required. While dysmorpholgy classification by expert clinical examination is not highly scalable, the use of automated tools for two and three-dimensional imaging[36] may make it feasible to perform high throughput dysmorphology classification. This will allow GRVSs to be more widely used, potentially in combination with one or more early clinical biomarkers.

## Methods

### Inclusion and ethics

We attempted to ensure that ethnic and other types of diversity in the research participants represented the populations being studied, including analyzing all family samples collected. We ensured that the study questionnaires were prepared in an inclusive way relevant to the populations being studied. For our ASD research, we also rely on participant advisory committees and our protocols undergo regular review. We designed our study to ensure sex balance in the recruitment of participants. For example, the male:female ratio for individuals with ASD in the Autism Speaks MSSNG (discovery) collection is similar the well-established 4:1 sex bias in the ASD population. All clinical and genomic data is available for further analysis (see main text and Supplementary Files), and the release of these data types is covered in the ethics protocols. The author list of this paper includes contributors from the locations where the research was conducted who participated in the data collection, design, analysis, and/or interpretation of the work. In the selection of authors, we followed guidelines of the *International Journal of Medical Education*. Each of these aspects of the study was part of the protocols approved by

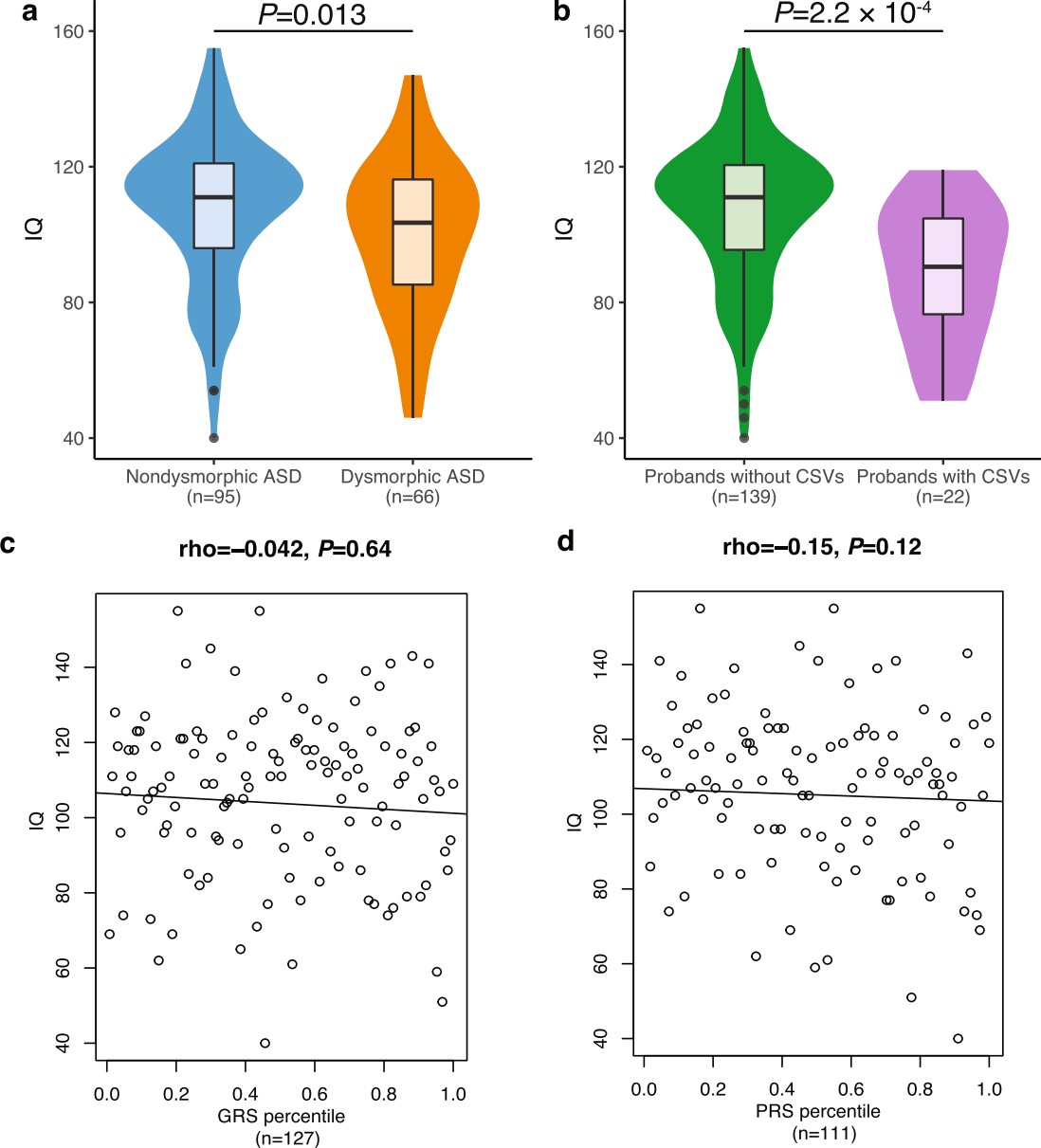

**Fig. 5 | Relationship between IQ, morphological ASD subtypes and genetic variants. a** Comparison of IQ among morphological ASD subtypes. **b** Comparison of IQ between probands with and without a CSV. **a, b** Violin plots show the distributions of the probands' IQ; box plots contained within show the median and quartiles of IQ for each subtype, the minima and maxima of box plots indicate 3× the interquartile range-deviated IQ from the median. *P* values denote the probability that the mean IQ of nondysmorphic ASD or probands without CSVs is not greater than dysmorphic ASD or probands with CSVs, respectively (one-sided, Welch's *t*-test). Correlation between IQ and (**c**) GRVS and (**d**) PRS is shown. **c, d** Each dot represents the IQ and GRVS or PRS percentile of a sample. The linear regression line indicates the linear correlation between IQ and GRVS or PRS percentiles. Correlation coefficient is quantified by two-sided Spearman's rho correlation. *P* values indicate the probability that the correlation is occurred due to chance. Source data are provided as a Source Data file.

Newfoundland's Health Research Ethics Board (HREB# 2003.027) and SickKids Research Ethics Board (REB#0019980189).

## Subject enrollment—discovery cohort

The cohort consists of children residing in the Canadian province of Newfoundland and Labrador, recruited from one of three developmental team assessment clinics between 2010 and 2018. Assessment through one of these clinics was required for a child with ASD to qualify for provincially funded home Applied Behavioural Analysis (ABA) therapy. Families were invited to participate after their child received an ASD diagnosis from the multidisciplinary team which was led by a developmental pediatrician. Probands met ASD criteria according to the Diagnostic and Statistical Manual of Mental Disorders (Fourth or Fifth Edition, Text Revision)[37,38] and all diagnoses were confirmed by an Autism Diagnostic Observation Schedule[39] assessment. Most probands also had an Autism Diagnostic Interview-Revised[40] assessment consistent with ASD. Children were not excluded from the study based on syndromic features or the presence of a known syndrome. Parents or guardians of the children provided written informed consent. The study was approved by Newfoundland's Health Research Ethics Boards (HREB# 2003.027) and SickKids Research Ethics Board (REB#0019980189).

## Subject enrollment—replication cohort

The replication cohort consisted of a subset of samples from the Simons Simplex Collection, including 442 affected probands with dysmorphology and WGS data along with their unaffected siblings (*n* = 355).

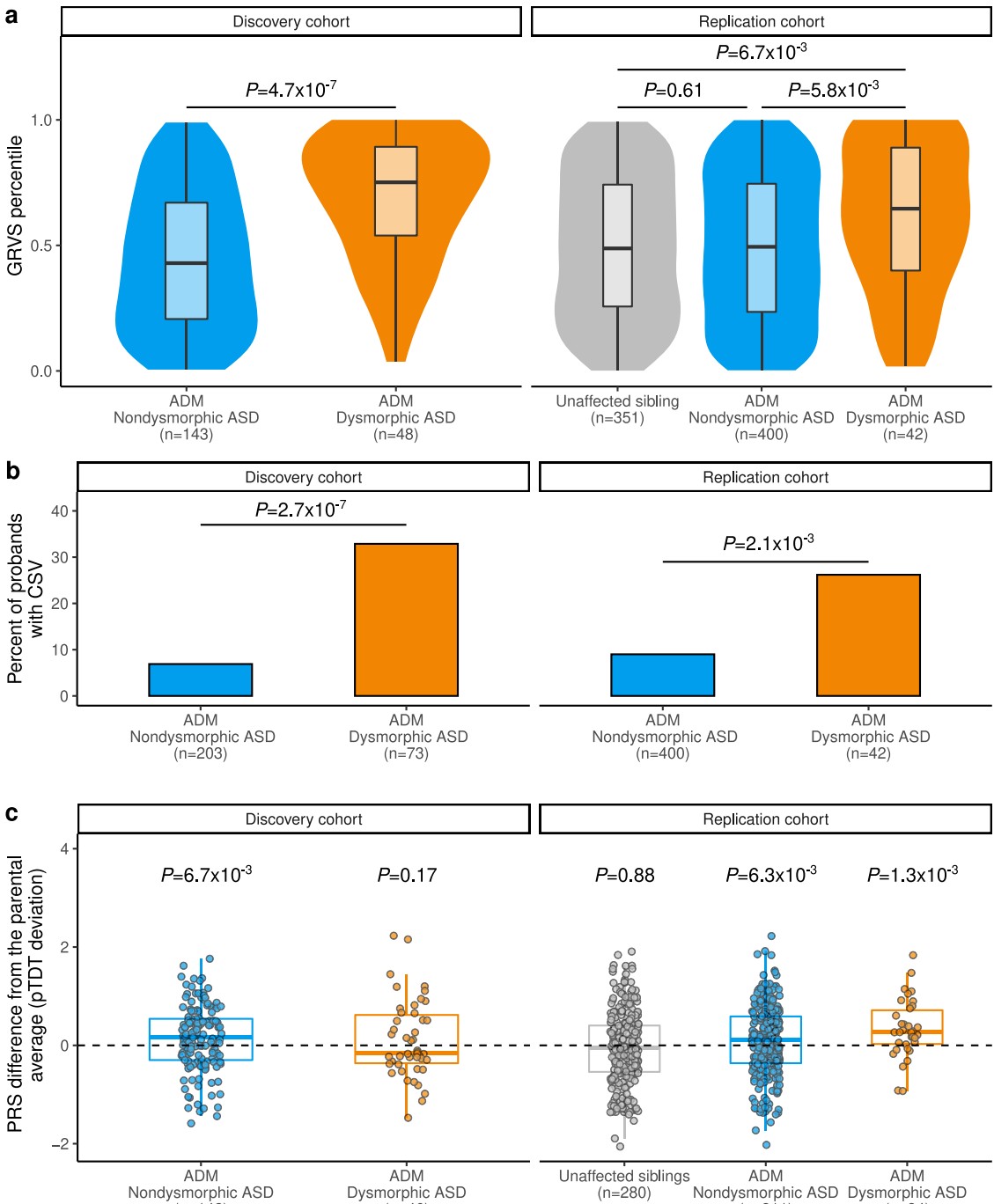

**Fig. 6 | Replication of rare and common genetic findings in a subset of Simons Simplex Collection cohort. a** GRVSs for each cohort were ordered and ranked by percentile. Violin plots show the distributions of the probands' GRVS percentiles; box plots contained within show the median and quartiles of the percentiles for each subtype, the minima and maxima of box plots indicate 3× the interquartile range-deviated scores from the median. $P$ values denote the probability that the GRVS in ADM-defined dysmorphic ASD is not greater than ADM-defined non-dysmorphic ASD (one-sided, Wilcoxon rank-sum test). **b** Yield of CSVs between dysmorphic and nondysmorphic subtypes in discovery and replication cohorts. $P$ values indicate the probability that the yield of CSVs in nondysmorphic ASD is not lower than that of dysmorphic ASD (one-sided, Welch's $t$-test). **c** Inheritance of polygenic risk for ASD in dysmorphic and nondysmorphic ASD subtypes in discovery and replication cohorts. Box plots depict the median and quartiles of pTDT deviation, and the minima and maxima of box plots indicate 3× the interquartile range-deviated pTDT deviations from the median. Dots represent pTDT deviations

of subjects. $P$ values for each subgroup indicate the probability that the mean of the pTDT deviation distribution is not greater than zero (one-sided, Welch's $t$-test), as depicted by the dotted line. The finding of no significant overtransmission in dysmorphic ASD did not replicate in SSC, which might be due to lack of statistical power (i.e., at least 100 dysmorphic samples are needed to achieve 80% power if PRS explains 2.45% of phenotypic variance[11]) and/or ascertainment differences between the discovery and replication cohorts. The discovery cohort included data about major congenital anomalies in morphologic classification, whereas the replication cohort did not. While our discovery cohort was population-based, the Simons Simplex Collection excluded probands with medically significant perinatal diseases, severe neurological deficits, and certain genetic syndromes[32]. This likely decreased the proportion of probands with excess MPAs and birth defects, potentially leading to a lower burden of common ASD-associated variants[88]. Source data are provided as a Source Data file.

## Clinical assessment and morphological examination–discovery cohort

Clinical assessments, morphological examinations, and classification were performed[9,10]. In brief, the team reviewed the child's family history and medical records, including radiology and electro-encephalogram (EEGs). EEGs were ordered if there was a clinical suspicion of seizures. Other screens for birth defects were arranged based on standard physical examination of the proband, which included a cardiovascular examination (e.g., echocardiogram for a proband with a murmur consistent with a ventricular septal defect). A single experienced dysmorphologist (B.A.F.) performed a detailed morphological examination of the child and (if possible) parents documenting minor physical anomalies (MPAs), height, weight, head circumference and anthropometric measurements of the head, face, hands, and feet. As described by Miles et al.[9], each proband was assigned an MPA score; one point was given for each embryologically unrelated MPA or for each measurement greater than two standard deviations above or below the population mean, and that was absent from the parents if they were available for examination. Each child was also assigned a major congenital anomaly (MCA) score (two points were given for each MCA), and a total morphology score (MPA + MCA scores). Using the total morphology score, we classified each child into essential (total morphology score 0–3), equivocal (total morphology score 4–5) or complex (total morphology score ≥6) groups. We used the final classification for comparing the yield of CSVs and for performing the rare and common variant analyses.

## Autism dysmorphology measure–discovery and replication cohorts

Our replication cohort consisted of a subset of samples from the Simons Simplex Collection[32]. This subset of samples had already been categorized into two morphological groups (400 nondysmorphic and 42 dysmorphic cases) by multiple non-geneticist examiners using the Autism Dysmorphology Measure[33]. In brief, the Autism Dysmorphology Measure is a decision tree-based classifier that assigns cases into nondysmorphic and dysmorphic groups based the presence or absence of minor physical anomalies of 12 body areas. It was designed to be used by clinicians who do not have expert training in dysmorphology, and the assessment is limited to the craniofacies, hands, and feet of the child. The ADM decision tree was trained on expert-derived consensus classification of 222 ASD cases who had gold-standard examinations of all body areas by clinical geneticists with expertise in dysmorphology[9,41]. The latter was the approach we used for the initial morphologic classification of our discovery cohort into essential, equivocal, and complex groups[9].

In contrast to the Autism Dysmorphology Measure, the morphological scores used to classify the discovery cohort factored in major congenital anomalies as well as MPAs, and MPAs were documented for the entire body including areas not assessed by the ADM (for example, the thorax, arms, legs, and skin). In order to align the type of morphologic data that was used to classify the discovery and replication cohorts, we reclassified the discovery cohort using the Autism Dysmorphology Measure, yielding 248 nondysmorphic and 77 dysmorphic cases. Of the 248 ADM-defined nondysmorphic cases, 18 cases were clearly dysmorphic upon further review by an experienced dysmorphologist (B.A.F). The Autism Dysmorphology Measure is reported to have an 82% sensitivity[33], and the sensitivity for the discovery cohort is similar at 80%. Thus, we excluded the 18 individuals with a false negative dysmorphic ADM classification to make the discovery cohort as clean as possible. We also included only samples that were sequenced on Illumina platforms to be consistent with the replication cohort[32]. Thus, the final number of ASD cases in the discovery cohort used for analysis was 276, of which 203 had nondysmorphic ASD and 73 had dysmorphic ASD, according to ADM.

## Whole-genome sequencing and variant detection

We extracted DNA from whole blood or lymphoblast-derived cell lines and assessed the DNA quality with PicoGreen™ and gel electrophoresis. We sequenced 795 genomes (325 probands and 470 parents) with one of the following WGS technologies/sites[4]: Complete Genomics (Mountain View, CA, n = 33 probands, 64 parents), Illumina HiSeq2000 by The Center for Applied Genomics (TCAG) (Toronto, ON, n = 24 probands, 48 parents), or Illumina HiSeq X by Macrogen (Seoul, South Korea, n = 182 probands, 250 parents) or TCAG (n = 86 probands, 108 parents). For WGS by Complete Genomics, at least 10 µg of non-degraded DNA was provided for WGS. Complete Genomics performed additional quality controls, including DNA quality assessment, sex check, and comparison of samples with results from 96-SNP genotyping assay to avoid sample mix-up. For WGS by Illumina HiSeq X, we used between 100 ng and 1 µg of genomic DNA for genomic library preparation and WGS. We quantified DNA samples using a Qubit High Sensitivity Assay and checked sample purity using the Nanodrop OD260/280 ratio. We used the Illumina TruSeq Nano DNA Library Prep Kit following the manufacturer's recommended protocol. In brief, we fragmented the DNA into 350-bp average lengths using sonication on a Covaris LE220 instrument. The fragmented DNA was end-repaired, A-tailed, and indexed using TruSeq Illumina adapters with overhang-T added to the DNA. We validated the libraries on a Bioanalyzer DNA High Sensitivity chip to check for size and absence of primer dimers and quantified them by qPCR using a Kapa Library Quantification Illumina/ABI Prism Kit protocol (KAPA Biosystems). We pooled the validated libraries in equimolar quantities and sequenced the paired-end reads of 150-bp lengths on an Illumina HiSeq X platform following Illumina's recommended protocol. For samples sequenced on HiSeq2000, DNA was extracted and sheared into fragments, which were then purified by gel electrophoresis. DNA fragments were ligated with adapter oligonucleotides to form paired-end DNA libraries with an insert size of 500 bp. We used ligation-mediated PCR amplification to enrich libraries with 5′ and 3′ adapters. The DNA libraries were sequenced to generate 90 bp pair-end reads with at least 30× average genome coverage per sample. We used KING v.2.2.5[42] to confirm familial relationships and ADMIXTURE v1.3[43] and EIGENSOFT v6.0beta[44] to confirm ancestries (Supplementary Data 12). PLINK version v1.9.b3.42 was used for basic QC and to format the input and results of both relatedness and ancestry analyses.

Alignment and variant calling for genomes sequenced by Complete Genomics were performed by Complete Genomics[45]. For samples sequenced on Illumina platforms, each WGS site aligned WGS reads to the human reference genome assembly hg19 (GRCh37) using Burrows-Wheeler Aligner v.0.7.12[46] (TCAG) or Isaac v.2.0.13[47] (Macrogen). For each genome, we performed local realignment and quality recalibration and detected SNVs and small indels using the Genome Analysis Toolkit (GATK) Haplotype Caller[48] v.3.4.6 without genotype refinement. We detected CNVs using ERDS (estimation by read depth with single-nucleotide variants)[49] 1.1 and CNVnator[50] 0.3.2. Algorithms were run using their default parameters. We used 500 bp as the window size for CNVnator. For CNVnator, we removed calls with >50% of q0 (zero mapping quality) reads within the CNV regions (q0 filter), except for the homozygous autosomal deletions or hemizygous X-linked deletions in males (with normalized average read depth; NRD < 0.03). We defined stringent calls as those that were called by both algorithms (with 50% overlap). In a subset of samples sequenced on HiSeq X, the insert size, $I$, fell below 350 bp, which correlated with fewer deletions called by ERDS. ERDS requires deletions <10 kb to be supported by anomalously mapped read pairs within a distance $D = I − 2r$, where $r$ is the read length. To resolve this, we modified ERDS that gives D a constant value of 50[51]. We detected SVs using Manta v.0.29.6[52]. When supported by the variant caller (i.e., GATK and Manta), trio-based joint variant calling was conducted for each family.

To identify uniparental isodisomies (isoUPDs), we calculated the ratio of the number of homozygous or hemizygous SNPs to the number of SNPs per chromosome, for each sample. Samples with a ratio greater than 0.55 had a putative isoUPD on the corresponding chromosome. We examined CNV and kinship data to rule out confounding factors (i.e., large CNVs or consanguinity). For each sample with a ratio greater than 0.55, we examined plots of B-allele frequency per chromosome; those with runs of homozygosity >10 Mb on one chromosome were considered to have a putative isoUPD[53]. We examined the inheritance of homozygous SNPs within the region of the putative isoUPD via visual inspection of BAM files and experimentally validated one of the SNPs to confirm the isoUPD and inheritance.

We systematically detected aneuploidies by calculating a ratio of the average read depth per chromosome to that for the entire sample. Ratios ≤0.5 and ≥1.5 were considered a loss or gain, respectively. For Complete Genomics data, we identified aneuploidies by looking for an excess of large CNVs for each chromosome per sample.

Tandem repeats were detected from samples with PCR-free DNA library preparation and sequenced on the Illumina HiSeq X platforms using ExpansionHunter Denovo[54] v0.7.0 with default parameters. We detected tandem repeat expansions in the discovery cohort using ExpansionHunter Denovo size cutoffs from the previous study[20]. Sample quality control procedures were performed to remove samples with tandem repeat counts exceed three standard deviations above mean[20].

## Variant annotation

We annotated SNVs and indels with information on population allele frequency, variant impact predictors, and putative pathogenicity and disease association, using a custom pipeline based on ANNOVAR Feb 2016 version[55] (see Supplementary Data 19 for list of databases and predictors used)[4]. For non-genic regions, we annotated whether the variant overlapped reported ASD-associated noncoding regions[21-25] (Supplementary Data 20). These included transcription start sites, fetal brain promoters and enhancers of LoF intolerant genes[22], histone modification (H3K27ac) sites in fetal and adult brain[23], splice sites, 3'- and 5'-untranslated regions (UTRs)[25], binding sites predicted by DeepSEA[24] to cause LoF, as well as conserved promoters of any genes, developmental delay-associated genes, and long noncoding RNA genes[21]. We tested three additional functional sites that have not been previously associated with ASD. These included boundaries of topologically associating domains[56], CTCF binding sites[57], and brain enhancers from Roadmap Epigenomics chromatin states (15-states chromHMM)[58].

We annotated CNVs and SVs with a custom pipeline using RefSeq gene models, with repeat regions, gaps, centromeres, telomeres and segmental duplications relative to the University of California at Santa Cruz genome assembly hg19. Similar to our non-genic annotations for SNVs, we annotated whether a CNV overlapped promoters of genes[21], H3K27ac sites[21], 3'UTR and 5'UTR[25] (Supplementary Data 20). We retained CNVs overlapping such regions, but not exonic regions. We also annotated the frequency of each CNV and SV from among 3107 parents in the MSSNG database[4] (fifth version) and the putative pathogenicity and disease association [from Human and Mouse Phenotype Ontologies[59,60] (HPO and MPO), ClinGen Genome Dosage Sensitivity Map[61], Online Mendelian Inheritance in Man, and Database of genomic variation and phenotype in humans using ensemble resources (DECIPHER)[62]].

We annotated mitochondrial variants using Annovar-based custom scripts with annotations from MitoMaster (April 2019) and Ensembl v96.

## Detection of rare variants

We extracted high-quality rare data for SNVs and indels after applying the following filters: (1) FILTER is PASS or varQuality is VQHIGH or PASS; (2) population allele frequencies <1% in 1000 Genome Project[63], NHLBI-ESP[64], Exome Aggregation Consortium[65], The Genome Aggregation Database[66], and internal Complete Genomics control databases; (3) reference and alternative allele frequency >95% and <1%, respectively, based on allele frequencies of 2573 parents in MSSNG (fourth version)[4] to decrease batch and cross-platform effects; and (4) allele frequency <5% from 250 parents from this study aligned with Isaac to decrease alignment-specific artifacts. The population allele frequency cutoff of <1% was selected, as it gave the optimal and significant analysis result (Supplementary Fig. 2). To further minimize cross-platform and batch effects, we required heterozygous SNVs and indels to have an alternative allele fraction of 0.3–0.7 (inclusive) and homozygous/hemizygous SNVs and indels to have an alternative allele fraction >0.7 for variants from Complete Genomics. For Illumina variants, we also required heterozygous SNVs and indels to have a genotype quality score of at least 99 and 90, respectively, and homozygous SNVs and indels to have a genotype quality score of at least 25.

We retained CNVs >2 kb that had <70% overlap with gaps, centromeres, telomeres, and segmental duplications. For CNVs from Illumina platforms, we defined stringent CNVs as those called by both ERDS and CNVnator (with 50% reciprocal overlap). We defined CNVs as rare if the allelic frequency was <1% in parents from the MSSNG database[4] and <5% in parents of this cohort that were aligned with Isaac.

We retained as rare SVs, those with an allelic frequency of <1% in parents analyzed with Manta from the MSSNG database and <5% in parents in this cohort that were aligned with Isaac. Pairs of entries with identical non-zero first numbers in the MATEID tag were retained as one inversion. Entries with identical MATEID values were retained as complex SVs. On average per sample, we detected ~3.7 million SNPs, 36,514 rare single-nucleotide variants (SNVs), 4113 small insertions and deletions (indels), 13 rare copy-number variants (CNVs), 390 rare structural variations (Supplementary Data 2).

## Detection of de novo variants

To identify de novo SNVs and indels from sequencing data from Complete Genomics, we compared each variant in the proband to the sequence at the same position in the parents. A variant inconsistent with Mendelian inheritance (present in the offspring but not in either parent or the sibling), was considered to be a potential de novo mutation for that child. We applied the following quality filters for each variant[45]: (i) varQuality of allele1 and allele2 is either VQHIGH (for v2.2) or PASS (for v2.4); (ii) ploidy of the child = 2 (or = 1 for X- and Y-linked variants in male subjects) and the ploidy of both parents is not "N"; (iii) the ratio of sequence reads supporting the alternative call to that of the reference call is 0.3–0.7 (or ≥0.7 for X- and Y-linked variants in male subjects); (iv) the variant call does not overlap with known regions of segmental duplication; (v) the refscore (likelihood of the region being the same as the reference sequence) in both parents is >40 or "−"; (vi) the variant call does not overlap with any variants found in Complete Genomics public genomes; (vii) the variant call has frequency <0.01 in the 1000 Genomes Project; (viii) the SNV call in the child does not overlap with any variant call (SNV or indel) in either parent; and (ix) variants clustered within a distance of 100 bp have been eliminated. For Illumina WGS data, we also used DenovoGear[67] (version 0.5.4) to detect de novo SNVs and indels. We extracted variants inconsistent with Mendelian inheritance (present in offspring but not in parents) with FILTER = PASS and defined rare, as above. To identify high-confidence de novo SNVs, we applied the following quality filters: (1) pp_DNM score ≥0.9 from DenovoGear[67]; (2) overlap GATK[48] calls with genotype quality scores ≥99 for heterozygous SNVs. We defined high-confidence de novo indels as those called by DenovoGear and GATK with the same start site. We retained de novo SNVs and indels with a ratio of sequenced reads supporting the alternative call to the total

number of reads at the position of 0.3–0.7, or >0.7 for X- and Y-linked variants not in the pseudoautosomal regions in male subjects.

We defined putative de novo CNVs as rare stringent CNVs (see "Detection of rare variants") that were inconsistent with Mendelian inheritance. For CNVs that did not have a conclusive inheritance pattern (i.e., CNV in child and parent were not the same size), we defined putative de novo CNV as those with a CNV length ratio between child and parent of >2. For each putative de novo CNV from Illumina platforms, we calculated a read depth ratio of the CNV with the surrounding region in each family member[51]. Ratios of 0.35–0.65 were considered heterozygous deletions, <0.35 as homozygous/hemizygous deletions, >=1.4 as duplications and 0.9–1.1 as a normal copy number. Putative de novo CNVs were considered de novo if the copy-number status based on ratios were inconsistent with Mendelian inheritance. For the 40 regions with ratios that did not meet the aforementioned criteria, we visualized the WGS reads to determine the inheritance status for samples sequenced by Illumina. To determine the inheritance status for samples sequenced by Complete Genomics, we examined the read depth coverage of the CNV relative to that of Complete Genomics controls[68] and its flanking regions in each family member. On average per sample, we detected 73.4 de novo SNVs, 7.3 de novo indels, and 0.1 de novo CNVs (Supplementary Data 2).

## Validation of variants

We randomly selected a subset of all high-quality exonic de novo SNVs, all de novo indels and all CSVs for validation in probands and available parents. We used Primer3[69] to design primers to span at least 100 bp upstream and downstream of a putative variant, avoiding regions of known SNPs, repetitive elements, and segmental duplications. DNA from whole blood, if available, was used to amplify candidate regions by polymerase chain reaction and to assay with Sanger Sequencing. For CNVs, we validated all high-confidence de novo exonic and all clinically significant CNVs in whole blood DNA (if available) of probands and available parents using TaqMan™ Copy Number Assay (Applied Biosystems), SYBR® Green qPCR (Thermofisher) or digital droplet PCR (BioRad). Experimental validation rates were 94.8%, 85.7%, and 87.5%, respectively, for de novo SNVs, indels, and CNVs (Supplementary Data 3 and 4).

## Mitochondrial variant detection

For the samples sequenced by Illumina platforms, reads aligning to the mitochondrial genome were extracted and realigned to the revised Cambridge Reference Sequence (NC_012920) in b37 using BWA v0.7.12. Pileups were generated with samtools mpileup v1.1 requiring the program to include duplicate reads in the analysis and retaining all positions in the output. Custom scripts were developed to parse the mpileup output to determine the most frequently occurring non-reference base at each position in the mitochondrial genome. The heteroplasmic fractions were calculated and vcf files were generated. Fasta files with the most frequently occurring base at every position were also generated and used as input for the program HaploGrep v2.1.1 for haplogoup prediction. The vcf files were annotated using Annovar-based custom scripts with annotations from MitoMaster (April 2019) and Ensembl v96.

For the samples that were sequenced by Complete Genomics, the mitochondrial variants called by the proprietary software were extracted. Fasta files were generated using custom scripts replacing mitochondrial reference bases with alternative bases at heteroplasmic sites, and the files were used as input for the program HaploGrep v2.1.1 for haplogoup prediction. The vcf files were annotated using Annovar-based custom scripts with annotations from MitoMaster (April 2019) and Ensembl v96.

Positions with heteroplasmic fraction less than 5% or greater than 95% and common in certain haplogroups (greater than 5%) were excluded from downstream analysis. All variants were manually reviewed, and a list of artefacts was compiled and excluded. To identify pathogenic mitochondrial variants, the following variants were considered: any MitoMaster pathogenic variants at 5–100% heteroplasmy, variants between 10–90% heteroplasmy, and variants between 5 and 100% heteroplasmy and seen <2% of the time in the individual's haplogroup.

## Variant detection for replication cohort

For the replication cohort, CRAM files and sequence-level variants were downloaded from Globus (https://www.globus.org/). We detected CNVs using ERDS[49] and CNVnator[50], as described for the discovery cohort[51]. Rare variants were filtered as described for the discovery cohort. We identified de novo SNVs and indels using DeNovoGear[67]. Allele frequencies from the Simons Simplex Collection were calculated and de novo variants with internal frequencies <1% were excluded. De novo SNVs and indels at poorly sequenced or highly variable sites were also excluded from further analysis. The remaining de novo variants were filtered as described for the discovery cohort, with the exception of using a PP_DNM < 0.95 threshold for de novo SNVs. Variants were annotated as described above for the discovery cohort.

## Variant prioritization and molecular diagnosis

To identify CSVs from the discovery cohort, we prioritized rare and de novo LoF and damaging (as predicted by at least five/seven predictors[25]) missense variants, and variants reported by ClinVar[70] or the Human Gene Variant Database[71]. We also prioritized rare and de novo CNVs and SVs, including those overlapping syndromic regions in DECIPHER[62] or ClinGen Genome Dosage Sensitivity Map[61] databases. Genes affected by such variants were compared to ASD candidate genes[3,4,13,72,73], candidate genes for neurodevelopmental disorders[72], and genes implicated in neurodevelopmental or behavioural phenotypes according to HPO[60] and MPO[59]. In addition, we considered the mode of inheritance from the Online Mendelian Inheritance in Man and Clinical Genomics Database[73], segregation and genotype–phenotype correlations. We classified the variants as pathogenic, likely pathogenic, variants of uncertain significance, likely benign, or benign, based on the American College of Medical Genetics and Genomics Guidelines[18,19]. Variants of unknown significance in known or candidate ASD genes with emerging evidence were further categorized into three ASD candidate variant categories (Supplementary Note 1 and Supplementary Data 5–7). Although applying quality filters for high-confidence variants is important to minimize false positives for burden analysis, this can increase false negatives. Therefore, we also manually inspected WGS data when we identified CSVs that did not pass filtering criteria for high-confidence variants.

Clinically significant variants classified as pathogenic or likely pathogenic or that were considered clinically relevant (i.e., prompting further clinical management) were reviewed by a medical geneticist in the context of the participant's phenotype and family history. Relevant findings were reported back to families through a clinical geneticist. Differences in the yield of CSVs among the morphological groups were calculated using Fisher's exact test.

To identify CSVs from the affected probands in the replication cohort, the aforementioned approach was applied to de novo LoF, damaging missense, and CNVs. CSVs from the replication cohort were confirmed by manual inspection of WGS reads.

## Rare variant burden analysis in gene sets and noncoding regions

For the discovery cohort, we performed two ASD subtype comparisons for each rare variant burden analysis as follows: (1) comparing complex, equivocal and essential ASD using ordinal regression tests and (2) comparing complex and equivocal ASD (i.e., dysmorphic ASD) to essential ASD using logistic regression tests. The test was done by regressing an event (e.g., number of genes impacted by rare deletions per subject) capturing a particular genomic region (i.e., coding, gene

sets, or noncoding regions) on the phenotype outcome (e.g., complex vs. essential ASD). The events tested in this study were the number of LoF, missense, and predicted deleterious variants for sequence-level variants and the number of genes or noncoding regions for CNVs. Tier 1 and 2 missense variants consist of all or only predicted damaging missense variants, respectively, as defined in ref. 25. The CNVs were grouped into two size bins, small CNVs (2–10 kb) and large CNVs (10 kb to 3 Mb) due to greater proportion of these CNVs overlapping coding or noncoding regions, respectively. The number of genes impacted by other CNVs was based on their overlap with the coding regions of each gene. However, the number of genes impacted by small CNVs were based on genic overlap since there were not enough small coding CNVs for the gene set enrichment analysis. We compiled a list of 37 gene sets related to neuronal function, brain expression, mouse phenotypes from MPO, or human phenotypes from HPO that have been previously associated with ASD or used as negative control gene sets when comparing ASD to control groups (Supplementary Data 21)[74–84]. For noncoding regions, we compiled a list of regions reported to be associated with ASD (Supplementary Data 20)[21–23]. We also included a score that predicts the impact of a variant on transcription factor binding as one of the noncoding regions tested[24]. Logistic regression and ordinal regression were applied for two subtypes and three subtypes comparison, respectively. Sex, genotyping platform, and three principal components from population stratification were included in the model as covariates to correct for any biases caused by sex difference, platforms, or ethnicity. Deviance test $P$ value was calculated by comparing residuals from two regression models; one with just the covariates and another with all both covariates and target variable[85]. Global burden analysis was performed to compare the total number of LOF variants, missense variants, predicted deleterious variants for sequence-level variants, and genes impacted by CNVs. The coefficients reported were obtained from the model with the covariates. Multiple test correction for global burden tests was done using the Benjamini Hochberg approach (BH-FDR). For the gene sets and noncoding regions burden test, total variant count (for SNVs and noncoding CNVs) or total gene count (for CNVs) was also included as one of the covariates to get rid of a global burden bias that might inflate the test $P$ value. The coefficients, however, were calculated from the model with all the covariates except the total variant count or the total gene count for the actual magnitude of their impact. Permutation-based FDR correction (1000 permutations) corrected for the multiple comparison. Since different gene sets and noncoding regions consist of the different number of genes or regions, we calculated the coefficients using z-scores for the number of features in each gene set/region to compare the coefficients across morphology-associated regions. When examining the burden of rare variants using logistic regression models, we used all probands from the discovery cohort ($n = 325$). Since some probands did not have their parents sequenced, we used a subset of the discovery cohort ($n = 235$) when examining de novo variants. All statistical analysis was performed using R Statistical Software v3.5.1.

## Genome-wide rare variant score

In addition to identifying relevant gene sets or regions that were differentially enriched among ASD morphologic subgroups, we developed a procedure to calculate a genome-wide rare variant score (GRVS) for each subject. This allowed the contribution of different variant types toward phenotype severity to be assessed together. The procedure involved two main steps: (i) identification of relevant, differentially enriched gene sets or noncoding regions for each variant type along with an estimation of their effect sizes in the discovery cohort, and (ii) calculation of the score for each subject in the target cohorts.

To estimate the effect sizes in the discovery cohort, we first fitted a logistic regression model by regressing platform, sex and first three

principal components from population stratification on the dysmorphology classification (nondysmorphic = 0 and dysmorphic = 1, or essential = 0, equivocal = 1, complex = 2). We then used the regression coefficients of these covariates and the intercept in the second logistic regression model, where a feature representing a particular gene set or region was tested. Therefore, regression coefficients of all the gene sets and regions were corrected for those possible biases from the covariates equally. The two models can be notated as below:

$$Y = a + \beta C \tag{1}$$

$$Y = a + \beta C + \beta_i X_i \tag{2}$$

where $Y$ is the outcome variable of dysmorphology classification, $a$ is an intercept, $\beta$ is a regression coefficients of covariates, $C$ is a vector of covariates, $\beta_i$ is the regression coefficient of a morphology-associated region, $i$, and $X_i$ is the number of features found in a morphology-associated region. A feature is defined as the number of rare or de novo SNVs or indels or the number of genes or noncoding regions impacted by rare CNVs. For rare variants, we used all probands in the discovery cohort. Since some probands did not have their parents sequenced, we used a subset of the discovery cohort when examining de novo variants. To determine the optimal $P$ value threshold to identify significant gene sets, we calculated Nagelkerke's $R^2$ at different $P$ value thresholds ($P < 0.001, 0.005, 0.01, 0.05, 0.1. 0.5,$ and $1$) using the discovery cohort and tenfold cross-validation strategy. The optimal $P$ value threshold was at $P < 0.1$ (Supplementary Fig. 4). To minimize the redundancy in significant gene sets and noncoding regions, we retained the most significant gene sets and noncoding regions with a Jaccard index <0.75. We used the regression coefficients ($\beta_i$) of significant gene sets or noncoding regions ($P < 0.1$) as a weight for the number of variants in those gene sets or regions in the GRVS calculation.

For each individual, the GRVS was calculated using the formula below

$$GRVS = \sum_{j=1}^{k} \sum_{i=1}^{n} \beta_{ij} X_{ij} \tag{3}$$

where $n$ is the number of significant ($P < 0.1$) gene sets or regions for a particular variant type, $j$, $k$ is the number of variant types (e.g., de novo missense variants), $\beta_i$ is a regression coefficient of a significant gene set or region, $i$, and $X_i$ is the number of variants (for SNVs and indels) or the number of genes or regions (for CNVs) that are found in the significant gene set or region in the sample.

To examine the GRVS in the discovery cohort, we used a tenfold cross-validation strategy to avoid overfitting. Using this strategy, the discovery cohort was randomly divided into 10 equally sized subsamples (stratified by subtypes). We calculated the GRVS of each sample in each subset using the effect sizes determined in the remaining nine subsets. To minimize stochasticity in the GRVS calculation, we repeated this procedure 30 times and the average GRVS and average number of variants for each sample were used for subsequent subtypes comparisons (Supplementary Fig. 1a). For the replication cohort, we calculated GRVSs using significant gene sets and effect sizes derived from the discovery cohort (Supplementary Fig. 1b). GRVS can be calculated for probands regardless of whether their parents have been sequenced. However, there would be a systematic difference in GRVSs in the discovery cohort if all probands were used because those whose parents have been sequenced includes scores from de novo variants, whereas probands whose parents have not been sequenced do not have scores from de novo variants. To ensure that the same variant types (including de novo variants) were included in each score for probands in the discovery cohort, GRVS was calculated only for

probands whose parents had also both been sequenced. GRVSs were standardized within each cohort and subtyping method. We tested whether GRVS is higher in dysmorphic ASD compared to non-dysmorphic ASD using a one-sided Wilcoxon's signed-ranked test.

We used our ADM-reclassified cohort as the discovery cohort for several reasons: (1) In contrast to the MPAs (dysmorphology data) from SSC which were identified by multiple non-geneticist examiners, MPAs in the discovery cohort were documented by a single dysmorphologist with over 20 years of clinical experience (B.A.F.). MPA's for children in the discovery cohort were then put through the ADM algorithm and the cases were classified as ADM-dysmorphic or ADM-nondysmorphic. This strategy allowed us to use very uniformly collected phenotypic data to derive the morphology-associated regions and effect sizes for GRVS calculation. (2) Our discovery cohort also contains more dysmorphic probands than SSC, which gives more power to identify morphology-associated regions (enriched in dysmorphic ASD). (3) Lastly, the discovery cohort was assembled using a population-based recruitment strategy so that the morphology-associated regions identified come from a patient collection representative of ASD as it exists at the level of primary care providers. In contrast there are ascertainment biases in SSC (e.g., simplex families and exclusion of severely affected/ syndromic probands) which might limit the generalizability of effect sizes and morphology-associated regions in a population-based cohort[35].

We calculated a score for CSVs using the GRVS formula if the CSV was identified in a proband with two sequenced parents, and if the variant occurred in or overlapped one of the morphology-associated gene sets or noncoding regions so that effect size was available for that variant. 46 CSVs were identified in 46 probands and17 of these met the above criteria allowing us to calculate a score for the variant. Of the remaining 29 CSVs, 15 were identified in probands where sequencing data were not available from both parents, and 14 variants did not overlap a morphology-associated region.

**Common variant and PRS analysis**
We examined the contribution of common SNPs among ASD subtypes. We calculated the PRS for each sample by deriving ASD summary statistics from a population-based genome-wide association study (GWAS) of 13,076 cases and 22,664 controls from the iPSYCH project[11]. We calculated the PRS for BMI, which was a negative control due to its lack of association with ASD[29], using BMI summary statistics from a population-based GWAS of 322,154 individuals of European descent from the GIANT Consortium[86]. We preprocessed the GWAS summary tables to fix the effect allele mismatch (swapped A1 and A2 alleles and converted its odds ratio) and to remove ambiguous SNPs (i.e., SNPs with A to T and C to G variations) and multi-allelic SNPs.

We conducted joint genotyping of BMI- and ASD-associated SNPs only on samples sequenced on Illumina platforms (200 probands and 400 parents). We could not re-genotype Complete Genomics data, so the samples were excluded from further analysis. We retained SNPs with a minor allele frequency >0.05 and genotyping rate >90%, of which 349,682 SNPs and 428,364 SNPs intersected with iPSYCH-ASD and GIANT-BMI SNPs passing suggested a $P$ value threshold ($P$ value < 0.1 for ASD and $P$ value < 0.2 for BMI) by Weiner et al.[2], respectively. We then calculated PRSs using PRSice v2.2.0[87] (parameters used: clump-kb 250, clump-p 1.000000, clump-r2 0.100000, info-base 0.9) using a $P$ value threshold of 0.1 for iPSYCH and 0.2 for GIANT-BMI, as suggested in ref. 8. After clumping, only 18,549 SNPs and 38,245 SNPs remained for PRS calculation for ASD and BMI, respectively[11]. We calculated PRS for ASD for the SSC replication cohort using 26,067 SNPs with $P$ value < 0.1 after the clumping step. The PRSs in both cohorts were standardized (with a mean of zero and standard deviation of one). We used the pTDT method[8] and one-sided Welch's $t$-test to examine the overtransmission of common variants associated with ASD susceptibility among subtypes. Probands were

used in the analysis if the probands were of European ancestry and if sequencing data were available from both parents.

**Reporting summary**
Further information on research design is available in the Nature Research Reporting Summary linked to this article.

## Data availability
The WGS data generated in this study have been deposited into controlled access research databases, as further described below, because this is the type of data sharing that was approved by the study participants. Access to FASTQ data for samples in the discovery cohort that were consented for MSSNG can be obtained by completing the data access agreement: https://research.mss.ng. Access to FASTQ data for samples in the discovery cohort not consented for MSSNG, as well as VCF files for sequence-level variants for all samples in the discovery cohort are available at European Genome-Phenome Archive (accession EGAS00001005753). This data can be obtained by contacting the corresponding author and completing the data access agreement. If approved, data will be shared through the European Genome-Phenome Archive. Access to data for the replication cohort can be obtained by completing the data access agreement (https://www.sfari.org/resource/sfari-base), as was done for this study. The clinical data generated in this study are provided in Supplementary Data 7, 12, and 16. Public databases used in this study can be accessed using the following links: 1000 Genomes Project (https://www.internationalgenome.org/), NHLBI Exome Sequencing Project (https://evs.gs.washington.edu/EVS/), gnomAD (https://gnomad.broadinstitute.org/), Human Phenotype Ontology (https://hpo.jax.org/app/), Mouse Phenotype Ontology (http://www.informatics.jax.org/vocab/mp_ontology), ClinVar (https://www.ncbi.nlm.nih.gov/clinvar/), Human Gene Mutation Database (https://www.hgmd.cf.ac.uk/), Clinical Genomics Database (https://research.nhgri.nih.gov/CGD/), and Online Mendelian Inheritance in Man (https://www.omim.org/). Source data are provided with this paper.

## Code availability
Code used in this manuscript is available at GitHub (https://doi.org/10.5281/zenodo.7113997).

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

## Acknowledgements

We thank the families for participation and The Centre for Applied Genomics for their analytical and technical support. We thank Lisa Strug, Andrew Paterson, and Delnaz Roshandel for analytical assistance. This work was funded by Autism Speaks (S.W.S.) Autism Speaks Canada (S.W.S.), the University of Toronto McLaughlin Centre (S.W.S.), the Canada Foundation for Innovation (S.W.S.), the Canadian Institutes of Health Research (CIHR) (S.W.S.), Genome Canada/Ontario Genomics Institute (S.W.S. and B.A.F.), the Government of Ontario (S.W.S.), Brain Canada (S.W.S.), The Swedish Foundation for Strategic Research (K.T., grant number: FFL18-0104), Ontario Brain Institute Province of Ontario Neurodevelopmental Disorders (POND) (E.A. and S.W.S.), and The Hospital for Sick Children Foundation (S.W.S.). A.J.S.C. was supported throughout this research by Ontario Graduate Scholarship from the Government of Ontario, Restracomp Research Fellowship from The Hospital of Sick Children, and Autism Research Training Award and Frederick Banting and Charles Best Scholarship from CIHR. S.W.S. holds the Northbridge Chair in Paediatric Research at the Hospital for Sick Children.

## Author contributions

A.J.S.C., R.K.C.Y., S.W.S., and B.A.F. conceived and designed experiments. B.A.F, C.N., N.T., and J.H.M. managed, recruited, diagnosed and examined participants. E.A., R.V.P., and J.V. helped with interpreting phenotype data. Z.W., B. Thiruvahindrapuram, B. Trost, T.N., G.P., W.S., and J.M. processed whole-genome sequencing data. A.J.S.C., W.E., R.K.C., D.M., and M.Z. conducted or interpreted different components of whole-genome sequencing analyses. A.J.S.C., M.S.R., D.J.S., N.S., and K.T. performed variant interpretation. A.J.S.C. and S.L. performed experiments for variant characterization and validation. A.J.S.C., W.E., B.A.F., J.H.L., and S.W.S. wrote the manuscript.

## Competing interests

At the time of this study and its publication, S.W.S. served on the Scientific Advisory Committee of Population Bio and was an Academic Consultant for the King Abdulaziz University. D.M. is a full-time employee and a shareholder of Deep Genomics Inc. D.J.S. has equity in PhenoTips. The remaining authors declare no competing interests.

## Additional information

[1]The Centre for Applied Genomics, Genetics and Genome Biology, The Hospital for Sick Children, Toronto, ON, Canada. [2]Genetics and Genome Biology, The Hospital for Sick Children, Toronto, ON, Canada. [3]CGEn, The Hospital for Sick Children, Toronto, ON, Canada. [4]Provincial Medical Genetics Program, Eastern Health, St. John's, NL, Canada. [5]Department of Psychiatry, The Hospital for Sick Children, Toronto, ON, Canada. [6]Department of Psychiatry, University of Toronto, Toronto, ON, Canada. [7]Department of Molecular Genetics, University of Toronto, Toronto, ON, Canada. [8]Division of Clinical and Metabolic Genetics, Department of Pediatrics, The Hospital for Sick Children, Toronto, ON, Canada. [9]Department of Pediatrics, University of Toronto, Toronto, ON, Canada. [10]Thompson Center for Autism and Neurodevelopmental Disorders, University of Missouri, Columbia, MO, USA. [11]Holland Bloorview Kids Rehabilitation Hospital, Toronto, ON, Canada. [12]Department of Women's and Children's Health, Karolinska Institutet, Stockholm, Sweden. [13]Deep Genomics Inc., Toronto, ON, Canada. [14]Department of Paediatric Laboratory Medicine, Genome Diagnostics, The Hospital for Sick Children, Toronto, ON, Canada. [15]Department of Laboratory Medicine and Pathobiology, University of Toronto, Toronto, ON, Canada. [16]Department of Pediatrics and The Saban Research Institute, Children's Hospital Los Angeles, Keck School of Medicine of University of Southern California, Los Angeles, CA, USA. [17]Discipline of Genetics, Faculty of Medicine, Memorial University of Newfoundland, St. John's, NL, Canada. [18]McLaughlin Centre, University of Toronto, Toronto, ON, Canada. ✉e-mail: bfernandez@chla.usc.edu; stephen.scherer@sickkids.ca

