## [Peer Review File · Nature Communications]

Genome-wide rare variant score associates with morphological subtypes of autism spectrum disorderReviewers' Comments:

Reviewer #1:

Remarks to the Author:

The study investigates clinical morphology data and whole-genome sequence data from a discovery cohort (325 ASD cases from MSSNG) and a replication cohort (442 ASD cases from SSC) to (i) develop a genome-wide rare variant score (GRVS) to measure the relationship between rare variants and morphology, and (ii) examine the contribution of rare and common variants in morphological ASD subtypes. The GRVS is a rare variant version of common variant polygenic score (PRS), here calculated as the sum of the number of rare variants in morphology-associated coding and non-coding regions, weighted by their effect sizes.

The authors find that cases with dysmorphic ASD have higher GRVS compared to those with nondysmorphic ASD in the discovery sample ($P=0.027$, one-sided). Perhaps more convincing results were obtained when assessing the less detailed Autism Dysmorphology Measure (ADM)-based morphology data available for the SSC replication sample (and aligned data in the discovery sample), showing higher GRVS in cases with ADM-defined dysmorphic ASD compared to ADM-defined nondysmorphic ASD in both cohorts ($P_{\text{discovery}}=3.6 \times 10^{-6}$ and $P_{\text{replication}}=2.7 \times 10^{-4}$, one-sided Wilcoxon rank sum test).

In addition, the authors report an over-transmission of ASD-associated common variants in nondysmorphic ASD probands in both the discovery and replication cohorts.

This is overall a well-conducted and thorough study. The phenotypic assessment of morphological subtypes is relevant and detailed, and, in spite of relatively modest sample sizes, the authors report interesting and replicable results. My major comments concern potential ancestry bias and selection of variants and gene sets:

- The rare variant burden analysis in gene sets and noncoding regions includes “three principal components from population stratification” as covariates. It is unclear how well this corrects for potential stratification due to ancestry. Also, how do the PCA plots compare for the different case subgroups? It is difficult to completely correct for potential ancestry bias in ancestrally mixed cohorts and it would seem appropriate to include a European-only analysis. This would seem relevant for both the burden analyses and the GRVS analyses.
- The authors include variants with population allele frequencies $< 1\%$ as rare. However, it is well-documented that the majority of the rare variant signal in ASD risk comes from much less frequent alleles, i.e. $MAF < 10^{-4}$ or 10^{-5} ; “ultrarare” variants. Thus, the rationale for a more inclusive threshold is unclear. As more frequent alleles would tend to introduce more noise than true signal it would seem relevant to present the results for such ultrarare variants (in burden, gene set enrichment and GRVS tests).
- For missense variants, the authors stratify in Tier 1 and 2 categories as defined in Yuen et al 2016. More recent studies find that variants with high MPC scores ($>2-3$) confer most of the ASD risk among missense variants and I was wondering if such classification would be more useful in the present study.
- Similarly, it would seem relevant to focus on LoF variants in constrained genes that have a low tolerance of LoF variants (assessed by e.g. gnomAD pLI or LOEUF scores) as that category carries most of the ASD risk.
- In the gene set analysis it would be informative to include the rare variant ASD risk genes identified in the most recent WES/WGS ASD studies (Satterstrom et al, Cell 2020, or even better Fu et al <https://www.medrxiv.org/content/10.1101/2021.12.20.21267194v1>), perhaps at 2-3 significance thresholds, and the 285 Bonferroni-significant genes from the DDD study (PMID: 33057194).

- The authors assess the influence of clinically significant variants (CSVs) on the GRVS analysis. It would also seem relevant to assess the impact of (ultra)rare deleterious LoFs in constrained genes (and large-effect missense variants, e.g. MPC>2 (and potentially CNVs)) that are robustly associated with ASD (e.g. the 71 FDR<0.001 associated genes in Fu et al, <https://www.medrxiv.org/content/10.1101/2021.12.20.21267194v1>) and/or in the 285 Bonferroni-significant genes from the DDD study.

Minor:

Suppl Table 9 says 36 gene sets while Suppl Table 20 and Methods mention 37 gene sets. Why this apparent difference?

Reviewer #2:

Remarks to the Author:

Chan et al. present a paper where they examine a discovery cohort of 325 individuals with ASD and a replication cohort of 442 individuals with ASD. In each cohort, they stratify individuals with autism by whether they are dysmorphic or nondysmorphic. The authors find higher rare variant burden in individuals with dysmorphic ASD and higher common variant burden in individuals with nondysmorphic ASD. The paper contains a small number of families. However, the findings in it are consistent with what is already known about autism. A few questions/comments below:

1. Since this study is quite small it would be important to frame it in the context of larger works (e.g., <https://www.medrxiv.org/content/10.1101/2021.03.30.21254657v1>). In particular, a comparison of the Antaki RVRS approach and the GRVS approach described here should be undertaken. It would also be important for understanding the novelty of this study.
2. What is the value of combining complex and equivocal ASD into dysmorphics ASD?
3. Are the 795 genomes in the discovery cohort new to this publication or have they been published in other papers by this group?
4. On page 17, what is TCAG?
5. Aneuploidies is spelled wrong in Figure 1 on page 42

Response to reviewer's comments (manuscript # NCOMMS-21-43407-T)

Reviewer #1:

1. **Reviewer:** The rare variant burden analysis in gene sets and noncoding regions includes “three principal components from population stratification” as covariates. It is unclear how well this corrects for potential stratification due to ancestry. Also, how do the PCA plots compare for the different case subgroups? It is difficult to completely correct for potential ancestry bias in ancestrally mixed cohorts and it would seem appropriate to include a European-only analysis. This would seem relevant for both the burden analyses and the GRVS analyses.

***Authors:** Opening general response to assist the reviewers: Any of our edits/changes discussed below are highlighted in yellow in the main text document. As discussed below, to help address the reviewer's comments we have added new Supplemental material (Investigation of minor allele frequency cut-off for rare variants and Population stratification and GRVS in European and non-European subsets sections) and edited display items (Figures 2, 3, 6, Supplementary Figures 2, 4-7, and Supplementary tables 9, 11-16, 20).*

Specific response to the first query: The discovery cohort consists of 317 individuals of European ancestry (97.5%) and the replication cohort consists of 655/797 individuals (including affected siblings) of European ancestry (82.2%). The PCA plots for the discovery cohort are shown in Supplementary Figure 7 and described in Supplementary Note. PCA data for replication cohort was not available. We did not find an association between different principal components and morphological subtypes as described in the Supplementary Note. As suggested by the reviewer, we compared GRVS between morphological subtypes using individuals only with European ancestry in the discovery and replication cohorts and found that the significant difference was retained. The replication cohort had enough samples of non-European ancestry for GRVS comparison between subgroups and unaffected siblings. We found that the significant results found in the European-only, and all ancestries analysis were retained in the non-European subset. Please see Supplementary Figure 7, Supplementary Note, and Lines 260-269 in the main text for more details.

2. **Reviewer:** The authors include variants with population allele frequencies $< 1\%$ as rare. However, it is well-documented that the majority of the rare variant signal in ASD risk comes from much less frequent alleles, i.e. $MAF < 10^{-4}$ or 10^{-5} ; “ultrarare” variants. Thus, the rationale for a more inclusive threshold is unclear. As more frequent alleles would tend to introduce more noise than true signal it would seem relevant to present the results for such ultrarare variants (in burden, gene set enrichment and GRVS tests).

***Authors:** This is an interesting avenue for further investigation. However, there may not be enough power in this study to examine ultra-rare variants. To address the reviewer's comments, we performed a burden analysis to identify the optimal allele frequency cut-off for rare variants in this study. Details of the methods and results are described in the*

supplementary information. We found that the optimal minor allele-frequency cut-off for this study was <1%.

- Reviewer:** For missense variants, the authors stratify in Tier 1 and 2 categories as defined in Yuen et al 2016. More recent studies find that variants with high MPC scores (>2-3) confer most of the ASD risk among missense variants and I was wondering if such classification would be more useful in the present study.

Authors: Thank you for the suggestion. We have used MPC in determining high impact variants as described in Lines 180-188 in the main text. We have also incorporated it into our GRVS analyses. There were not enough missense variants with MPC >2 in each of the genesets listed in Supplementary Table 20 in both cohorts. Thus, we grouped missense variants with MPC >2 together as one set when calculating GRVS.

- Reviewer:** Similarly, it would seem relevant to focus on LoF variants in constrained genes that have a low tolerance of LoF variants (assessed by e.g. gnomAD pLI or LOEUF scores) as that category carries most of the ASD risk.

Authors: We have included LoF intolerant genes to this study as reflected in Sup Table 20, and results in Sup table 11, 14, and 15.

- Reviewer:** In the gene set analysis it would be informative to include the rare variant ASD risk genes identified in the most recent WES/WGS ASD studies (Satterstrom et al, Cell 2020, or even better Fu et al <https://www.medrxiv.org/content/10.1101/2021.12.20.21267194v1>), perhaps at 2-3 significance thresholds, and the 285 Bonferroni-significant genes from the DDD study (PMID: 33057194).

Authors: We have included the Fu et al. and Kaplanis et al. genesets into our study as reflected in Supplementary Table 20, and results in Supplementary Table 11 and 14. We also added reference to the Fu et al. medrxiv and Kaplanis et al. papers.

- Reviewer:** The authors assess the influence of clinically significant variants (CSVs) on the GRVS analysis. It would also seem relevant to assess the impact of (ultra)rare deleterious LoFs in constrained genes (and large-effect missense variants, e.g. MPC>2 (and potentially CNVs)) that are robustly associated with ASD (e.g. the 71 FDR<0.001 associated genes in Fu et al, <https://www.medrxiv.org/content/10.1101/2021.12.20.21267194v1>) and/or in the 285 Bonferroni-significant genes from the DDD study.

Authors: Based on our reply to comment #2 above, this study would not have enough power to investigate ultra-rare variants. Instead, we assessed the impact of rare (<1%) high impact (i.e., LoF or missense variants with MPC > 2) in ASD-associated genes in Fu et al. on GRVS. We removed the samples with these rare high impact variants in 183 ASD-associated genes and compared the GRVS of the remaining samples between morphological subgroups and found that those with dysmorphic ASD still had a higher

non-significant ($P=0.07$) average GRVS than those with nondysmorphic ASD. Please refer to lines 180-188 in the main text.

7. **Reviewer:** Suppl Table 9 says 36 gene sets while Suppl Table 20 and Methods mention 37 gene sets. Why this apparent difference?

Authors: We apologize for the typo. The number of genesets mentioned in the title of Supplementary Table 9 has been updated to fix the typo and include additional genesets suggested by the reviewer.

Reviewer #2:

1. **Reviewer:** Since this study is quite small it would be important to frame it in the context of larger works (e.g., <https://www.medrxiv.org/content/10.1101/2021.03.30.21254657v1>). In particular, a comparison of the Antaki RVRS approach and the GRVS approach described here should be undertaken. It would also be important for understanding the novelty of this study.

Authors: We added to page 4 to address the above. The novelty of our study is that it elaborates the rare variant score further by weighing the variants by gene sets defined by function, expression, or disease association, in addition to weighing the variants by variant type (the genomic and phenotypic data is also novel to this study, both collected by our own group). This should help increase specificity and reduce noise when calculating the score. In addition, our study includes both sequence and copy number variants in the GRVS, whereas the afore-mentioned paper only includes sequence variants in the RVRS. We have also added reference to the now peer-reviewed study by Antaki et al., which has just been published.

2. **Reviewer:** What is the value of combining complex and equivocal ASD into dysmorphic ASD?

Authors: Thank you. We combined complex and equivocal ASD into dysmorphic ASD to increase power in the discovery cohort (explanation added on page 5). We also combined complex and equivocal ASD into dysmorphic ASD to make the phenotype groups more comparable between discovery and replication cohorts.

3. **Reviewer:** Are the 795 genomes in the discovery cohort new to this publication or have they been published in other papers by this group?

Authors: Yes, the 795 genomes in the discovery cohort are new to this publication and lines 115-116 have been modified. In addition to the data previously described as available in the MSSNG resource, since the initial submission of the manuscript, VCF files for sequence-level variants are available at European Genome-Phenome Archive (<https://ega-archive.org/studies/EGAS00001005753>).

4. **Reviewer:** On page 17, what is TCAG?

***Authors:** TCAG stands for The Centre for Applied Genomics, the 1998-founded genome centre at the Hospital for Sick Children (lead site of this study), which is defined in the first paragraph on page 17 when describing the WGS sites and technologies. An important aspect of our study is the original high-quality phenotype and genome sequence data (the latter from TCAG).*

5. **Reviewer:** Aneuploidies is spelled wrong in Figure 1 on page 42

***Authors:** We have corrected this typo in Figure 1.*

Reviewers' Comments:

Reviewer #1:

Remarks to the Author:

The authors have addressed most of my comments satisfactorily and I'm happy to support publication of the revised manuscript.